# Large Language Models for Biomedical Knowledge Graph Construction

## Abstract

The automatic construction of knowledge graphs (KGs) is an important research area in medicine, with far-reaching applications spanning drug discovery and clinical trial design. These applications hinge on the accurate identification of interactions among medical and biological entities. In this study, we propose an end-to-end machine learning solution based on large language models (LLMs) that utilize electronic medical record notes to construct KGs. The entities used in the KG construction process are diseases, factors, treatments, as well as manifestations that coexist with the patient while experiencing the disease. Given the critical need for high-quality performance in medical applications, we embark on a comprehensive assessment of 12 LLMs of various architectures, evaluating their performance and safety attributes. To gauge the quantitative efficacy of our approach by assessing both precision and recall, we manually annotate a dataset provided by the Macula and Retina Institute. We also assess the qualitative performance of LLMs, such as the ability to generate structured outputs or the tendency to hallucinate. The results illustrate that in contrast to encoder-only and encoder-decoder, decoder-only LLMs require further investigation. Additionally, we provide guided prompt design to utilize such LLMs. The application of the proposed methodology is demonstrated on age-related macular degeneration.

## 1 Introduction

There are several biomedical data corpora available that provide valuable knowledge, and one such source is PubMed Kilicoglu et al. (2012). PubMed is a search engine that accesses MEDLINE Kilicoglu et al. (2012), which is a database of abstracts of medical publications and references. Moreover, the widespread adoption of electronic medical records (EMR) has brought various opportunities for medical knowledge discovery. Knowledge graphs (KG) are often used for knowledge discovery, because graph-based abstraction offers numerous benefits when compared with traditional representations. They have been applied to various areas of healthcare, including identifying protein functions Santos et al. (2022), drug repurposing Drancé et al. (2021), and detecting healthcare misinformation Cui et al. (2020). Another application may be a clinical trial design Skelly et al. (2012), during which identification of confounding variables is an important step. Confounding variables may mask an actual association, or, more commonly falsely demonstrate an apparent association between the treatment and outcome when no real association between them exists.

KGs are a powerful tool for organizing and representing knowledge in a graph structure, where nodes represent entities within a specific domain, while edges symbolize relationships between these entities. The type of relationships may vary depending on the domain, allowing for the use of directed or undirected graphs. For example, in Nordon et al. (2019), they employed a directed graph to encode causal relationships between diseases. Other KGs may utilize both symmetric and asymmetric relationships. In our work, we specifically focus on using directed graphs to represent relationships between diseases and various factors, treatments, and manifestations that coexist with a patient while experiencing the disease (referred to as 'coexists_with').

Recent advancements in large language models (LLM) offer an opportunity to think about their ability to learn valuable representations from the knowledge encoded in medical corpora. Effectively analyzing textual data and KG construction requires extensive domain knowledge and is often a

time-consuming process for medical experts. To address this challenge, we propose an end-to-end method for automatically constructing knowledge graphs from electronic medical record (EMR) notes using LLMs, specifically through relation extraction.

Previous studies have suggested the utilization of specific LLMs for clinical relation extraction Agrawal et al. (2022); Sushil et al. (2022). However, due to the inherent safety-critical nature of healthcare, we conducted a comprehensive analysis of the performance and safety attributes of LLMs with varying architectures. To evaluate and assess their potential for medical applications and to address potential safety concerns, we introduced a manually annotated, private dataset and benchmarked the performance of 12 distinct LLMs. We have not performed an analysis on publicly available EMR datasets, such as MIMIC-III Johnson et al. (2016), because some of the models have used these datasets for training or fine-tuning. Our analysis revealed that in contrast with encoder-only and encoder-decoder models, decoder-only models need further guidance to output in a structured manner, which is required for relation extraction to construct the KG. We, therefore, introduced a guided prompt design that helped to utilize some of such LLMs for our task and analyzed issues that are making others unsuitable. This rigorous assessment forms a critical foundation for the safe and effective deployment of LLMs in the healthcare domain. Our work takes the form of the following contributions:

- We present a end-to-end method leveraging LLMs for the automatic construction of KGs from EMR notes

- We conduct an extensive and rigorous evaluation of the performance of 12 LLMs of various architectures specifically tailored for clinical relation extraction

- We provide guided prompt design to utilize decoder-only LLMs for relation extraction to construct KG between aforementioned medical entities

## 2 RELATED WORK

One notable success in the construction of knowledge bases (KBs) from biomedical textual data is SemRep Rindflesch & Fiszman (2003). SemRep is a rule-based system that combines syntax and semantics with biomedical domain knowledge contained in the Unified Medical Language System (UMLS) Bodenreider (2004) for semantic relation extraction. The range of predicates in SemRep is diverse, including molecular interactions, disease etiology, and static relations.

SemRep operates through a pipeline of five steps. Firstly, pre-linguistic analysis is performed, which includes sentence splitting, tokenization, and acronym detection. SemRep heavily relies on MetaMap Aronson (2001) for this step. Secondly, lexical/syntactic analysis is performed, which involves part-of-speech tagging, subcategorization, and grammatical number identification. SemRep heavily relies on the UMLS SPECIALIST Lexicon Lu et al. (2020) for this step. Thirdly, referential analysis identifies named entities and maps them to ontological concepts. Fourthly, post-referential analysis filters out semantically empty words/phrases and establishes semantic dependencies between noun phrases (NP). Lastly, relational analysis generates predications based on lexical, semantic, and syntactic knowledge. Shalit et al. Nordon et al. (2019) further improve the precision of SemRep by adding three additional filtration steps.

As one may observe, SemRep utilizes various levels of language modeling. It has been experimentally demonstrated that LLMs intrinsically learn these levels of language specification, without explicit programming Søgaard (2021). In Sung et al. (2021), BERT-based models with probing are used to extract relations between biomedical entities. The authors observe that, although LLMs can extract biomedical knowledge, they are biased towards frequently occurring entities present in prompts. We do not argue about the bias of LLMs, but rather the complexity of extracting relations via probing. We propose providing larger context information than that which is solely present in the prompt.

Trajanoska et al. (2023) makes connection between LLMs and semantic reasoning to automatically generate a Knowledge Graph on the topic of sustainability. It further populates it with concrete instances using news articles from the internet. It experiments with REBEL Huguet Cabot & Navigli (2021) and ChatGPT and shows that ChatGPT OpenAI (2023) is able to automatically create KGs from unstructured text, if reinforced with detailed instructions.

The paper on few-shot clinical extraction using LLMs Agrawal et al. (2022) discusses the challenge of extracting important variables from clinical data and presents an approach that leverages large language models, specifically InstructGPT Ouyang et al. (2022), for zero-shot and few-shot information extraction from clinical text. The authors demonstrate the effectiveness of this approach in several NLP tasks that require structured outputs, such as span identification, token-level sequence classification, and relation extraction. To evaluate the performance of the system, the authors introduce new datasets based on a manual reannotation of the CASI dataset Moon et al. (2014).

We argue that our setup is more complex as we do not consider clean, well-written, academic corpora such as PubMed Kilicoglu et al. (2012) and CASI Moon et al. (2014). The EMR corpus contains a significant amount of grammatical errors ("there is some heme OD .. ?"). Practitioners use abbreviations and notations ("RTO") not defined in the context, obfuscating the underlying information even further. Our study benchmarks different LLMs of varying architectures and training procedures on this challenging dataset.

## 3 DATASET

For this cohort study, data was obtained from the EMR of the Macula & Retina Institute, an independent health system in Glendale, California, USA. The dataset included approximately 10,000 patient records of individuals with retina-related eye diseases who had visited the institute between 2008 and 2023. The study focused on extracting knowledge from the clinical notes, which are records of observations, plans, and other activities related to patient care. These notes contain a patient's medical history and reasoning and can be used to identify complex disease-related patterns such as potential treatments, causes, and symptoms. In total, the study analyzed 360,000 notes relating to 122 unique eye diseases.

### 3.1 DATASET PREPROCESSING

Clinical notes frequently have repetitive segments conforming to a standardized template utilized by medical practitioners. The preprocessing step primarily involved computing the cosine similarity between note pairs. If the similarity score exceeds threshold (denoted threshold_preprocessing, more in Appendix F), the one with a higher word count is prioritized to preserve more informative content. Additionally, notes comprising fewer than 5 words are excluded from further analysis.

## 4 PROPOSED METHOD

Our proposed approach constructs a KG of diseases and their factors, treatments, and manifestations that the patient exhibits while undergoing the disease. To achieve this, the system initially identifies disease-specific notes as described in Subsection 4.1. Next, for each category of medical entity, we design set of questions (Subsection 4.5). We leverage an LLM to answer a pre-designed set of questions, taking into consideration the aforementioned disease-specific notes as contexts as described in Subsections 4.3 and 4.6. The list of LLMs that we experimented with are available in Subsection 4.2. The Subsection 4.7 discuss postprocessing techniques utilized to get the final relations to construct the KG.

### 4.1 DISEASE-SPECIFIC NOTES IDENTIFICATION

In clinical records, a single disease, denoted as $c_{input}$, may be expressed in multiple ways. The set of such expressions is denoted as $C$. These expressions may vary between clinics as well. To identify all instances of $c_{input}$ in the records, we employ the Unified Medical Language System (UMLS) Metathesaurus Bodenreider (2004), a comprehensive repository of biomedical terminologies and ontologies containing over 3 million concepts and their corresponding aliases, such as diseases, drugs, and procedures. We first check if any of the expressions in $c_i \in C$ appear in the record, and if so, we add the record to a list of disease-specific records for $c_{input}$. Moreover, clinicians may make typographical errors when recording the condition in the notes. To account for this, we use the BioBERT NER model to extract a list of diseases, denoted as $C_{note}$, present in the record. We then calculate the cosine similarity between each expression $c_{note_i} \in C_{note}$ and $c_{input}$. If the similarity is

above threshold (denoted threshold_notes_identification, more in Appendix F), we add the record to the result list. Refer to Appendix C.1 for more details on the algorithm.

## 4.2 MODELS

| Architecture | Model | Size | PTT |
|---|---|---|---|
| Encoder-only | BioBERT-SQuAD-v2 | 110M | 137B |
| | BERT-SQuAD-v2 | 110M | 137B |
| | RoBERTa-SQuAD-v2 | 125M | 2.2T |
| Decoder-only | BioGPT | 349M | - |
| | OPT | 30B | 180B |
| | OPT-IML-MAX | 30B | 180B |
| | Llama 2 | 70B | 2T |
| | Vicuna | 33B | 2T |
| | BLOOM | 176B | 366B |
| | WizardLM | 70B | 2T |
| Encoder-decoder | FLAN-T5-XXL | 11B | 34B |
| | FLAN-UL2 | 20B | 1T |

Table 1: We show all the models used in this paper, as well as their size, architecture and the number of pretraining tokens. We focus only on pretraining data, and ignore any finetuning data. PTT stands for pretraining tokens.

Table 1 shows all the models that we used in this paper. Our main objective revolves around experimenting with various architectures of LLMs and analyzing their performance through a comprehensive evaluation that brings forward potential edge cases and safety attributes. To accomplish this, we conducted experiments using different LLM models categorized under three architectures: encoder-only, decoder-only, and encoder-decoder. Our next objective was to include as much diverse LLMs as possible encompassing variations in size as well as the number of pretraining tokens. For more detailed insights into each individual model, please refer to Appendix A.

## 4.3 ALIGNING LLMS FOR RELATION EXTRACTION

In this work, we assume only query access to a large language model (i.e., no gradients). The task is to identify relations by finding answers to specific queries. We explore two distinct approaches for aligning large language models to the task: open-book QA Gholami & Noori (2021) and in-context learning Brown et al. (2020). QA tries to find an answer to a given query. In the case of open-book QA, a query consists of a question and a context. An LLM tries to find an answer to the question from the context. the context serves as a specific, external information source from which the model is expected to extract answers to the posed questions. The context is typically a passage, document, or a set of information separate from the model's pre-existing knowledge. The model's task is to directly reference and pull information from this provided context to respond to the query. Thus, the context in open-book QA acts as a discrete reference material that the model consults to find answers. While it is commonly associated with encoder-only language models, in-context learning is specifically tailored for decoder-only and encoder-decoder models. For in-context learning, we give the LLM a prompt that consists of a list of input-output pairs that demonstrate a task. While in-context learning also involves a question and a context, the usage and purpose of the context differ significantly. Here, the context is not an external source of information but an integral part of the model's prompt. It includes a series of input-output pairs that serve as examples to guide the model in understanding the nature of the task. These examples demonstrate how to process and respond to similar queries. Therefore, the context in in-context learning is instructional and part of the learning material embedded within the prompt, guiding the model's response generation process based on demonstrated patterns, rather than serving as a source from which to extract direct answers. In this

study we focused on zero-shot Wei et al. (2022), few-shot Brown et al. (2020), and instruction-based prompting Ye et al. (2023).

## 4.4 PROMPT DESIGN

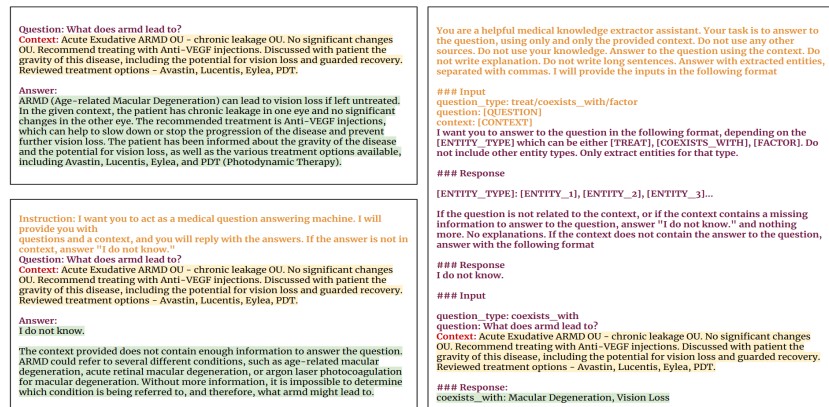

Figure 1: Guided instruction-based prompting. Each prompt design element is denoted by specific annotations. In the top left, a simple prompt structure includes a retrieval message and sentence introduction. The bottom left showcases an instruction prompt, featuring an overall task instruction and retrieval message. On the right, guided instruction prompt is detailed, containing overall task instructions, a retrieval message, and specific input-output formatting instructions. Different colors indicate different prompt design elements: orange for overall task instructions, red for sentence introduction, purple for the retrieval message portion and green for the LLM response

We conduct a comparison of three prompting techniques: zero-shot, few-shot, and instruction-based. In the case of zero-shot and few-shot approaches, we simply append entity-related questions to the input. Additionally, for the few-shot approach, we provide an example input/output.

Regarding the instruction-based prompting, we follow a systematic and task-agnostic process to construct prompts as outlined in Jimenez Gutierrez et al. (2022). As depicted in the left examples in Figure 1, this method identifies three key components of a prompt: overall task instructions, a sentence introduction, and a retrieval message. Building upon this, we introduce guided instruction-based prompting, denoted as 'guided' in the forthcoming sections. This refined prompt structure incorporates three fundamental elements:

- **Contextual Task Instruction:** This aspect furnishes explicit and comprehensive guidance, emphasizing the model's role in extracting information solely from the provided context. It establishes a clear framework for understanding the task at hand.

- **Input Format Guidance:** To mitigate issues of relations extracted out of context, we introduced explicit instructions on how the model should interpret and process input. This includes specifying the acceptable types and formats for questions and contexts.

- **Output Format Guidance:** We refined the retrieval message to include the entity type and explicitly instruct the model on the desired format for its responses.

For a detailed visualization of guided prompt structure and its components, please refer to Figure 1.

## 4.5 QUESTION DESIGN

We define template questions like "What treats %s". The "%s" in the questions represents a place-holder for a disease. All the predicates (e.g. treats, affect, cause, factor) are taken from SemRep Rindflesch & Fiszman (2003). The questions are categorized into three types: treatment-related, factor-related, and coexists_with-related questions. The treatment-related questions inquire about methods to slow down the progression, decrease the chance, or reduce the risk of a specific condition.

The factor-related questions aim to identify the causes, factors, or risks associated with a condition. The coexists_with-related questions explore any symptoms, effects, diseases, clinical tests, or behaviors that may manifest in the patient while experiencing the disease. The full list of questions for the LLM queries is available in Appendix B.

## 4.6 RELATION EXTRACTION

Our relation extraction method uses an LLM to answer a question given a related context. The design of the questions is described in Subsection 4.5. Questions are defined relative to a disease $d$ from the set of diseases $D$ and denoted as $q(d)$ by replacing the placeholder of the question with $d$. A context of a disease $d$ is a clinical note that contains $d$. The set of all clinical notes identified by Subsection 4.1 for a disease $d$ is denoted by $C(d)$. The set of questions of the category (e.g. treatment) $t$ is denoted by $Q_t(d)$ for a disease $d$. Algorithm 1 describes the process of querying the LLM for diseases and clinical notes with the defined questions.

---

**Algorithm 1** Querying LLM

---

**Ensure:** $result$
  $result := \{\}$
  **for** $d \in D$ **do**
    **for** $t \in \{treatment, factor, coexists\_with\}$ **do**
      **for** $q(d) \in Q_t(d)$ **do**
        $tmp := \langle LM(\langle c, q(d) \rangle), d, t \rangle$         ▷ Where LM returns a list of possible answers
                                           ▷ with their probabilities.
        $result.insert(tmp)$
      **end for**
    **end for**
  **end for**

---

The list of scores collected from Algorithm 1 is denoted as $result$. One may notice that entries of $result$ are triplets where the first element is a list. The $result$ is expanded by its first dimension. For example, if $result = \{\langle \{a, b\}, d, t \rangle\}$ then it becomes $result = \{\langle a, d, t \rangle, \langle b, d, t \rangle\}$ after the expansion. A single probability estimate may be unreliable for a relation of type $t$ between $e$ and $d$. This is strongly highlighted in Nordon et al. (2019) where text methods exhibit low precision by overestimating the probabilities for untrue relations. Motivated by this, we average over multiple occurrences of the relation of the type $t$ between $e$ and $d$. We then keep the relations that have occurred more than relation_occurrence_number times and that have an average probability higher than relation_probability (more on relation_occurrence_number and relation_probability in Appendix F). The numbers are chosen arbitrarily and they may be tuned for a dataset. Finally, the relation with the highest probability is chosen as the final prediction for an ordered pair of entities. Refer to Appendix C.2 for more details.

## 4.7 POSTPROCESSING

To map the model's output to a list of values for each medical entity, we initially filtered out the predictions with a probability score lower than 0.08. Subsequently, to remove meaningless information, stop words and punctuation were excised from each predicted text.

In addition to these steps, our approach included handling instances where the model explicitly expressed uncertainty or lack of sufficient context. In scenarios where the large language model (LLM) responds with variations of "I do not know" due to insufficient or ambiguous context, these responses were identified and systematically filtered out. This is crucial, as it ensures that our knowledge graph is built only on reliable, contextually supported information rather than on uncertain or speculative model outputs.

Further analysis revealed that models tend to generate the same answers in various forms depending on the given context. For instance, predictions such as "areds" and "areds-2 vitamins" essentially refer to the same value for a specific medical entity, but are expressed differently. To address these variations, we employed normalized cosine similarity for the tokens in the model's predictions. Specifically, for each medical entity, we calculated the cosine similarity between each pair of predictions. Predictions

with a similarity score exceeding 0.8 were considered equivalent and subsequently grouped together. From each group, the prediction with the highest initial probability score assigned by the model was selected. Finally, the refined output was converted into a list of values, selecting spans of text directly from the LLM output. A qualitative example illustrating this process is provided in Appendix D.

## 5 RESULTS

We now describe our experimental study over our techniques for constructing the KG.

**Setup** We construct a KG for age-related macula degeneration (AMD), which is a progressive eye disease mainly occurring in older people with a high incidence rate.

The absence of absolute ground truth leave us to compare the results with the knowledge of medical practitioners, which may be incomplete. Moreover, new knowledge is likely to be discovered when using large textual and EMR repositories Nordon et al. (2019). To evaluate the LLMs, we needed to review all clinical notes related to AMD and extract all factors, treatments and coexists_with terms. The evaluation is done based on precision and recall, with the groundtruth for comparison being the entity values available in the notes. We refer to this extraction process as an annotation. This annotation was carried out by two of the authors, a retina specialist, and a clinical research coordinator. To establish a consistent annotation schema, a set of examples was jointly annotated. Following this, each annotator independently annotated the same set of examples, and the two sets of annotations were then combined via a joint manual adjudication process. The AMD related notes have been identified according to Subsection 4.1 and preprocessed as described in Subsection 3.1. These steps leave us with 320 clinical notes.

| Architecture | Model | Treatment | | Factor | | Coexists_with | |
|---|---|---|---|---|---|---|---|
| | | Precision | Recall | Precision | Recall | Precision | Recall |
| Encoder-only | RoBERTa-SQuAD-v2 | 0.25 | 0.54 | 0.21 | 0.75 | 0.3 | 0.14 |
| | BioBERT-SQuAD-v2 | 0.13 | 0.9 | 0.25 | 0.75 | 0.45 | 0.71 |
| | BERT-SQuAD-v2 | 0.17 | 0.45 | 0.17 | 0.45 | 0.17 | 0.57 |
| Encoder-decoder | FLAN-T5-XXL: 0-shot | 0.55 | 0.75 | 0.54 | 0.69 | 0.64 | 0.89 |
| | FLAN-T5-XXL: few-shot | 0.45 | 0.9 | 0.66 | 0.8 | 0.72 | 0.88 |
| | FLAN-T5-XXL: instruct | 0.86 | 0.9 | 0.8 | 0.8 | 0.83 | 0.97 |
| | FLAN-T5-XXL: guided | 0.88 | **1** | 0.82 | **0.875** | 0.76 | 0.875 |
| | FLAN-UL2: 0-shot | 0.43 | 0.9 | 0.16 | 0.62 | 0.74 | 0.85 |
| | FLAN-UL2: few-shot | 0.55 | 0.9 | 0.36 | 0.75 | 0.78 | 0.89 |
| | FLAN-UL2: instruct | **0.98** | **1** | 0.8 | 0.8 | **0.98** | **1** |
| | FLAN-UL2: guided | **0.98** | **1** | **0.84** | **0.875** | **0.98** | **1** |
| Decoder-only | Vicuna-33B: guided | 0.63 | **1** | 0.5 | 0.75 | 0.46 | 0.75 |
| | Llama-2-70B: guided | 0.65 | **1** | 0.38 | 0.75 | 0.4 | 0.875 |
| | WizardLM-70B: guided | 0.78 | **1** | 0.61 | **0.875** | 0.5 | 0.875 |

Table 2: We are comparing the performance of LLMs with various architectures across all three medical entities. The evaluation is based on precision and recall measurements for each medical entity within the final KG. The baseline for comparison are the entity values available in the notes. 'guided' refers to the guided instruction-based prompting described in Subsection 4.4.

**Precision and recall results**

Table 2 shows the precision and recall results of different LLMs of various architectures. The best performance is consistently achieved with encoder-decoder LLMs for most medical entities. Specifically, FLAN-UL2, when used with instruction-based prompting, outperforms the other models. Furthermore, we observe that encoder-decoder models using 0-shot and few-shot prompting techniques are comparable to encoder-only and decoder-only models in some cases; however, when instruction-based prompting is employed, they significantly outperform the others.

Quantitative results for decoder-only models using 0-shot, few-shot, and instruction-based prompting techniques are not available. These models did not produce structured outputs, rendering them unsuitable for our task. Additional information can be found in Decoder-only models.

Unlike other prompting techniques, guided instruction-based prompting (as described in Subsection 4.4), has shown significant improvements. This advancement has enabled us to utilize three decoder-only models for this task out of the seven we experimented with. These models are Llama 2 Touvron et al. (2023), Vicuna-33B Zheng et al. (2023), and WizardLM-70B Xu et al. (2023). Notably, WizardLM-70B achieves the highest recall for factors and treatments, demonstrating that the incorporation of additional guidance has enhanced the understanding of the task by some of the decoder-only models, resulting in more precise and accurate answers. We believe that further research is required to explore the potential of decoder-only models for challenging relation extraction tasks, and future investigations may enhance their reliability. See prediction examples in Appendix E.

**Decoder-only models**

Here we describe the challenges that make some of these models unsuitable for clinical relation extraction, thus KG construction. Some of the models are prone to "hallucinating", a term commonly used to refer to the models generating responses that are factually incorrect or nonsensical. See such examples in Appendix E.2.1.

Furthermore, we observed cases where some models generated correct responses, but these responses did not originate from the given context. Another concern was the generation of excessively verbose or repetitive responses. Despite being contextually correct, the lengthy and redundant nature of these outputs complicated the postprocessing phase, making the integration of such responses into our KG construction pipeline impossible. See such examples in Appendix E.2.2.

**Qualitative Example: AMD**

We continue using AMD as a qualitative example. AMD is a progressive eye disease affecting the retina, specifically the macula. The risk factors for AMD have been studied extensively and have widely been known to include age, race, smoking status, diet, and genetics Holz et al. (2014); Heesterbeek et al. (2020). The exact reasons and mechanisms behind AMD are not yet fully researched. There are multiple pathways and factors for drusen formation and AMD progression, so it is hard to disentangle them. Large and numerous drusen are associated with an increased risk of developing advanced AMD Schlanitz et al. (2019). The pathophysiologic landscape of AMD potentially involves degenerative transformations within several ocular components, including the outer retinal layers, the photoreceptors, retinal pigment epithelium (RPE) characterized by the loss of the ellipsoid zone (EZ) and atrophic changes, accumulation of subretinal/submacular fluids, perturbations in Bruch's membrane leading to choroidal neovascularization (CNVM), and areas of choriocapillaris nonperfusion resulting in macular atrophy and fibrosis Holz et al. (2014); Boyer et al. (2017). Medical evaluators annotated drusen, genetics, CNVM, smoking, RPE irregularities, submacular/subretinal fluid, fibrosis, and loss of EZ zone as risk factors for AMD. The KG constructed with the utilization of FLAN-UL2 with instruction-based prompting that have relatively the best quantitative performance, is visually presented in Table 3.

| Treatment | Factor | Coexists_with |
|---|---|---|
| AREDS vitamins | Drusen | Poor visual acuity |
| Avastin | Genetics / Family history | Metamorphopsia |
| Lucentis | Peripheral CNVM/CNVM | Visual changes |
| PDT | Smoking | Macula Risk genetic testing |
| WACS vitamins | RPE irregularity | Wet AMD |
| Amsler grid testing | Submacular fibrosis and fluid | Dry AMD/GA |
| Spinach | Loss of EZ zone | ForeseeHome |
| Fish | Diabetes | Drusen |
| Omega-3 fatty acids | Glaucoma | Amblyopia |
| Anti-VEGF | Subretinal fluid | |
| Green Leafy Vegetables | | |
| Lack drusen | | |

Table 3: KG for AMD constructed using FLAN-UL2 model with instruction-based prompting. Red color indicates an incorrect values. Orange color indicates a values missed by the model.

Notably, besides factors, the graph also highlights a spectrum of terms that are linked to potential treatments and symptoms associated with AMD. Among the treatment entities are AREDS/WACS vitamins, dietary interventions, and Anti-VEGF treatments including Avastin and Lucentis. Other treatments indicated include PDT (Photodynamic Therapy), the utilization of Amsler grid, supplementation of Omega-3 fatty acids, and consumption of specific foods such as fish, spinach, and green leafy vegetables. The symptomatic aspects of AMD encompass a range of visual impairments and clinical manifestations. Patients afflicted with AMD often experience poor visual acuity, metamorphopsia (distorted vision), and can be diagnosed with either dry or wet AMD. Additionally, the management of the condition often involves undergoing assessments such as ForeseeHome and Macula Risk genetic testing, which play a pivotal role in monitoring the progression and development of AMD. Each of these terms is identified as values to the 'Coexists_with' entity within the graph.

| Treatment | Factor | Coexists_with |
| --- | --- | --- |
| Injection procedure | Blind Vision | Visual impairment |
| Photochemotherapy | Antioxidants | Massive hemorrhage |
| Antioxidants | Oxidative Stress | Autofluorescence |
| Bevacizumab | | Blindness |
| Eye care | | Legal, Disability NOS |
| Homocysteine thiolactone | | |
| Operative Surgical Procedures | | |

Table 4: KG for AMD constructed using SemMedDB.

We also show the KG constructed by SemMedDB Kilicoglu et al. (2012) in Table 4. SemMedDB is a repository of semantic predictions extracted from the titles and abstracts of all PubMed citations. It is evident that our approach has identified terms not found in the SemMedDB. Our method may not forge new terms where none existed in the original medical literature repository. However, the feedback from our medical evaluators underscores its potential to contribute to novel discoveries by highlighting existing but overlooked information.

## 6    CONCLUSION

In this paper, we propose an end-to-end approach that harnesses LLMs for the automatic generation of KGs from EMR notes. KGs hold significant value in numerous healthcare domains, including drug discovery and clinical trial design. The entities involved in the KG construction process encompass diseases, factors, treatments, and manifestations that co-occur with patients experiencing these diseases. Through extensive evaluation across various LLM architectures, we have demonstrated that encoder-decoder models outperform others in clinical relation extraction. Furthermore, we emphasize the need for additional investigation into the suitability of decoder-only models for medical applications, particularly given their critical safety implications. We believe that an automated knowledge extraction method may deliver substantial benefits to the medical community and facilitate further research in the field.

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

# APPENDIX

## A  MODELS

**Encoder-only models**

Our approach utilizes a fine-tuned question-answering model based on BERT Devlin et al. (2019), specifically fine-tuned on the SQuAD v2 dataset Rajpurkar et al. (2016). This model, which we refer to as BERT-SQuAD-v2, benefits from the core principles of BERT, including random token masking during pretraining to encourage contextual understanding.

Inspired by advancements in the BERT family, we also incorporate RoBERTa Liu et al. (2019), which is improved upon Bert by introducing a new pretraining recipe that includes training for longer and on larger batches, randomly masking tokens at each epoch instead of just once during preprocessing, and removing the next-sentence prediction objective. We also consider BioBERT Lee et al. (2020), which is a pre-trained BERT model which is trained on different combinations of general & biomedical domain corpora.

**Decoder-only models**

BioGPT Luo et al. (2022), a generative Transformer model tailored for biomedical literature, has shown remarkable results on several biomedical NLP benchmarks, including an impressive 78.2% accuracy on PubMedQA Jin et al. (2019). However, our efforts to employ BioGPT for relation extraction were met with challenges. The model frequently hallucinated during inference, making it unsuitable for our specific application in relation extraction.

Open Pretrained Transformers (OPT) Zhang et al. (2022) represents a comprehensive suite of decoder-only transformers designed for large-scale research. OPT-30B, a particular model from this suite, has been pre-trained predominantly on English text with some multilingual data from CommonCrawl. Sharing similarities with GPT-3, it uses a causal language modeling (CLM) objective. OPT-IML Iyer et al. (2022) represents an advanced version of the OPT model, enhanced with Instruction Meta-Learning. It's been trained on an extensive collection known as the OPT-IML Bench, comprising roughly 2000 NLP tasks from 8 different benchmarks. Two variations exist: the standard OPT-IML trained on 1500 tasks, and OPT-IML-Max that covers all 2000 tasks.

BLOOM Scao et al. (2022) stands as a sophisticated autoregressive Large Language Model (LLM), designed to produce coherent text across 46 languages and 13 programming languages, replicating human-like text generation capabilities.

Llama 2 Touvron et al. (2023) is a distinguished collection of generative text models, with models ranging from 7 billion to 70 billion parameters. Presented by Meta, this repository encompasses the 70B variant, made compatible with the Hugging Face Transformers framework. Within the Llama 2 family lies a specialized series called Llama-2-70B-Chat, meticulously fine-tuned for dialogue-centric applications. This model excels, outstripping many open-source chat models in benchmarks and rivalling prominent closed-source counterparts like ChatGPT and PaLM in terms of helpfulness and safety.

Emerging from the wave of advanced chatbots, Vicuna-33B Zheng et al. (2023) stands out as an open-source contribution, fine-tuned using the LLaMA framework based on dialogues from ShareGPT. Notably, when evaluated using GPT-4, Vicuna-33B not only showcased a commendable performance, rivaling the likes of OpenAI's ChatGPT and Google Bard (achieving over 90%* quality), but also surpassed counterparts like LLaMA and Stanford Alpaca Taori et al. (2023) in over 90%* of the tests. This exceptional achievement comes at a modest training cost of around $300, making Vicuna-33B an attractive proposition. Additionally, its code, weights, and a live demo are accessible for the research community, albeit restricted to non-commercial applications.

WizardLM-70B Xu et al. (2023) is a Large Language Model (LLM) built on the foundation of LLaMA, incorporating a novel training approach known as Evol-Instruct. This method involves leveraging artificial intelligence to evolve complex instruction data, setting WizardLM apart from LLaMA-based LLMs trained on simpler instructions. As a result it outperforms counterparts in tasks that demand intricate understanding and execution of instructions.

**Encoder-decoder models**

FLAN-T5-XXL Chung et al. (2022) is a encoder-decoder model that has been pre-trained on a multi-task mixture of unsupervised and supervised tasks and for which each task is converted into a text-to-text format. It performs well on multiple tasks including question answering.

FLAN-UL2 Yitayew (2023) is an encoder-decoder model based on the T5 architecture. It uses the same configuration as the UL2 Tay et al. (2022) model released earlier last year and was fine-tuned using the "Flan" prompt tuning and dataset collection Wei et al. (2022). According to the original blog, there are some notable improvements over the original UL2 model. The Flan-UL2 checkpoint uses a receptive field of 2048 which makes it more usable for few-shot in-context learning. This Flan-UL2 checkpoint does not require mode tokens anymore.

In comparison to FLAN-T5, FLAN-UL2 outperforms FLAN-T5 XXL on all four setups with an overall decent performance lift of +3.2% relative improvement. Most of the gains seem to come from the CoT setup while performance on direct prompting (MMLU and BBH) seems to be modest at best.

## B  QUESTION LIST

| Medical entity | Question |
| --- | --- |
| Treatment | What can slow the progression of %s? (T1) |
| | What can decrease the chance of %s? (T2) |
| | What can reduce the risk of %s? (T3) |
| | What is a treatment for %s? (T4) |
| | What treats %s? (T5) |
| Factor | What does cause %s? (F1) |
| | What is the cause of %s? (F2) |
| | What is the factor for %s? (F3) |
| | What can increase the risk of %s? (F4) |
| | What can convert to %s? (F5) |
| Effect | What can %s convert to? (E1) |
| | What is the effect of %s? (E2) |
| | What does %s lead to? (E3) |
| | What can %s become? (E4) |
| | What does %s affect? (E5) |

Table 5: List of questions categorized by the medical entity. The "%s" in the questions represents a placeholder for a disease.

## C ALGORITHMS

### C.1 DISEASE-SPECIFIC NOTES IDENTIFICATION

---

**Algorithm 2** Disease-specific notes identification

---

**Ensure:** $result$
  $result := \{\}$
  $C$ = UMLS_Metathesaurus_API($c\_input$)
  **for** $note$ in $clinical\_notes$ **do**
    $C_{note}$ = BioBERT_NER($note$)
    **if** $note$ contains $c_i$ **then**
      $result$.append($note$)
    **else**
      $similarity\_score$ := calculate_cosine_similarity($c_{input}, c_{note_i}$)
      **if** $similarity\_score > threshold$ **then**
        $result$.append($note$)
      **end if**
    **end if**
  **end for**

---

### C.2 RELATION EXTRACTION

---

**Algorithm 3** Relation extraction

---

**Require:** $result$ from Algorithm 1
  $relation := \{\}$
**Ensure:** relations
  **for** unique $\langle e, d, t \rangle$ in $result$ **do**
    $temp := \langle average(result[e, d, t].score), count(result[e, d, t]) \rangle$
    **if** $temp.average \geq 0.1$ and $temp.count \geq 10$ **then**
      $stat \leftarrow \langle temp.average, e, d, t \rangle$
    **end if**
  **end for**
  **for** unique $e, d$ in $stat$ **do**
    $relations \leftarrow \langle d, \arg\max_t stat[e, d], e \rangle$
  **end for**

---

# D POSTPROCESSING

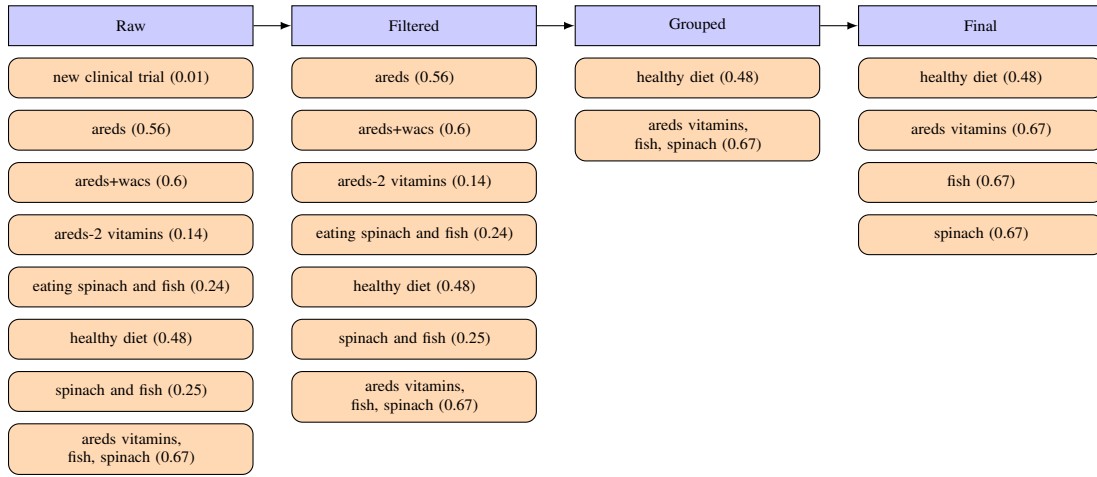

Figure 2: Qualitative example of the postprocessing steps. Every orange node illustrates the predictions made by an LLM, along with an associated probability enclosed in parentheses.

# E PROMPTS AND SAMPLE OUTPUTS

## E.1 ENCODER-ONLY MODELS

### E.1.1 EXAMPLES OF WRONG PREDICTIONS

Listing 1: BERT-SQuAD-v2: wrong prediction

```
Question: What can slow the progression of macular degeneration?
Context: Macular Degeneration: Discussed the nature of dry macular
    degeneration. Discussed Age Related Eye Disease Study and recommended
    AREDs vitamins for prevention purposes. Patient given Amsler grid to
    monitor for metamorphopsias or changes in central vision.
Answer:
dry macular degeneration
```

Listing 2: RoBERTa-SQuAD-v2: wrong prediction

```
Question: What does cause Macular Degeneration?
Context: Macular Degeneration: Discussed the nature of dry macular
    degeneration. Discussed Age Related Eye Disease Study and recommended
    AREDs vitamins for prevention purposes. Patient given Amsler grid to
    monitor for metamorphopsias or changes in central vision.
Answer:
dry macular degeneration
```

### E.1.2 EXAMPLES OF RIGHT PREDICTIONS

Listing 3: BERT-SQuAD-v2: right prediction

```
Question: What can slow the progression of myopic macular degeneration?
Context: Myopic Macular Degeneration - Explained that there is no
    specific treatment at this time. AREDS and WACS vitamins MAY help
    slow down the progression of the degeneration. Monitor closely. All
    questions were answered to the patient's satisfaction.
Answer:
AREDS and WACS vitamins
```

Listing 4: BioBERT-SQuAD-v2: right prediction

```
Question: What does cause ARMD?
Context: Macular Degeneration: Discussed the nature of dry macular
    degeneration. Patient encouraged to use an Amsler grid to monitor
    macular function by looking for metamorphopsias or visual changes.
Answer:
metamorphopsias or visual changes
```

Listing 5: RoBERTa-SQuAD-v2: right prediction

```
Question: What does cause amd?
Context: Smoking is the largest 'modifiable' risk factor for AMD - by
    stopping smoking a patient can reduce the chance of getting AMD and
    of AMD progressing.There is a very strong association between smoking
     and macular disease. Smokers are up to four times more likely than
    non-smokers to develop age-related macular degeneration (AMD) and
    smokers with a genetic predisposition to AMD are eight times more
    likely to get the condition. Some research suggests people with
    certain genes can be 20 times more likely to get macular degeneration
     if they smoke.
Answer:
Smoking
```

## E.2 DECODER-ONLY MODELS

### E.2.1 EXAMPLES OF HALLUCINATIONS

Listing 6: BLOOM with instruction-based prompting: hallucination

```
I want you to act as a question answering machine. I will provide you
    with a questions and a context and you will reply with the answers.
Question: What can slow the progression of armd?
Instruction: If the answer is not in context answer "i do not know".
Context: Wet ARMD- Patient educated on condition. Discussed diet and use
    of Macular Degeneration vitamins. All questions were answered to the
    patient's satisfaction.
Answer:
i do not know

    a:
    i think the question is a bit vague.  it is not clear what the
    question is about.  it is not clear what the context is.  it is not
    clear what the answers are.  it is not clear what the question is
    asking.  it is not clear what the question is asking the answerer to
    do.  it is not clear what the question is asking the answerer to do.
    it is not clear what the question is asking the answerer to do.  it
    is not clear what the
```

Listing 7: BLOOM with guided instruction-based prompting: hallucination

```
You are a helpful medical knowledge extractor assistant. Your task is to
    answer to the question, using only and only the provided context. Do
    not use any other sources. Do not use your knowledge. Answer to the
    question using the context. Do not write explanation. Do not write
    long sentences. Answer with extracted entities, separated with commas.
     I will provide the inputs in the following format

### Input
question_type: treat/coexists_with/factor
question: [QUESTION]
context: [CONTEXT]

I want you to answer to the question in the following format, depending
    on the [ENTITY_TYPE] which can be either [TREAT], [COEXISTS_WITH], [
    FACTOR]. Do not include other entity types. Do not include treatment
    if not asked. Only extract entities for that type.
```

```
### Response

[ENTITY_TYPE]: [ENTITY_1], [ENTITY_2], [ENTITY_3]...

If the question is not related to the context, or if the context contains
     a missing information to answer to the question, answer "I do not
    know." and nothing more. No explanations. If the context does not
    contain the answer to the question, answer with the following format

### Response
I do not know.

### Input

question_type: coexists_with
question: What does armd lead to?
context: Acute Exudative ARMD OU - chronic leakage OU. No significant
    changes OU. Recommend treating with Anti-VEGF injections. Discussed
    with patient the gravity of this disease, including the potential for
     vision loss and guarded recovery. Reviewed treatment options -
    Avastin, Lucentis, Eylea, PDT.

### Response
effect:
    If the question contains multiple entities, extract all of them,
    separated with commas.  For example, if you have following question
    Question 1:
```

Listing 8: BioGPT with guided instruction-based prompting: hallucination

```
You are a helpful medical knowledge extractor assistant. Your task is to
    answer to the question, using only and only the provided context. Do
    not use any other sources. Do not use your knowledge. Answer to the
    question using the context. Do not write explanation. Do not write
    long sentences. Answer with extracted entities, separated with commas.
     I will provide the inputs in the following format

### Input
question_type: treat/coexists_with/factor
question: [QUESTION]
context: [CONTEXT]

I want you to answer to the question in the following format, depending
    on the [ENTITY_TYPE] which can be either [TREAT], [COEXISTS_WITH], [
    FACTOR]. Do not include other entity types. Do not include treatment
    if not asked. Only extract entities for that type.

### Response

[ENTITY_TYPE]: [ENTITY_1], [ENTITY_2], [ENTITY_3]...

If the question is not related to the context, or if the context contains
     a missing information to answer to the question, answer "I do not
    know." and nothing more. No explanations. If the context does not
    contain the answer to the question, answer with the following format

### Response
I do not know.

### Input

question_type: factor
```

```
question: What is the factor for armd?
context: 2 small Druse OD- clinically does not look like ARMD. Patient
    has a family history of ARMD, recommend starting on AREDS + WACS
    vitamins. Eat green leafy vegetables like Spinach 5 times a week and
    fish at least 2 times a week.

### Response
I do not include any other entities to answer the question.
```

Listing 9: OPT-30B with guided instruction-based prompting: hallucination

```
You are a helpful medical knowledge extractor assistant. Your task is to
    answer to the question, using only and only the provided context. Do
    not use any other sources. Do not use your knowledge. Answer to the
    question using the context. Do not write explanation. Do not write
    long sentences. Answer with extracted entities, separated with commas.
     I will provide the inputs in the following format

### Input
question_type: treat/coexists_with/factor
question: [QUESTION]
context: [CONTEXT]

I want you to answer to the question in the following format, depending
    on the [ENTITY_TYPE] which can be either [TREAT], [COEXISTS_WITH], [
    FACTOR]. Do not include other entity types. Do not include treatment
    if not asked. Only extract entities for that type.

### Response

[ENTITY_TYPE]: [ENTITY_1], [ENTITY_2], [ENTITY_3]...

If the question is not related to the context, or if the context contains
     a missing information to answer to the question, answer "I do not
    know." and nothing more. No explanations. If the context does not
    contain the answer to the question, answer with the following format

### Response
I do not know.

### Input

question_type: coexists_with
question: What can amd convert to?
context: Explained to patient that he does have AMD but it is mild and
    not the cause of his blurred vision.

### Response
effect:  [ENTITY_1], [ENTITY_2], [ENTITY_3]...
    If the question is not related to the context, or if the context
    contains a missing information to answer to the question, answer Ï do
    not know.änd nothing more.  No explanations.  If the context does not
    contain the answer to the question, answer with the following format
    ### Response
    I do not know.
    ### Input
    question_type:  factor
    question:  What can amd convert to?
    context:  Explained to patient that he does have AMD but it is mild
    and not the cause of his blurred vision.
    ### Response
    factor:  [ENTITY_1], [ENTITY_2], [ENTITY_3]...
```

```
If the question is not related to the context, or if the context
contains a missing information to answer to the question, answer Ï do
not know.änd nothing more.  No explanations.  If the context does not
contain the answer to the question, answer with the following format

### Response

I do not know.

### Input

question_type:  treat

question:  What can amd convert to?

context:  Explained to patient that he does have AMD but it is mild
and not the cause of his blurred vision.

### Response

treat:  [ENTITY_1], [ENTITY_2], [ENTITY_3]...

If the question is not related to the context, or if the context
contains a missing information to answer to the question, answer Ï do
not know.änd nothing more.

No explanations.

If the context does not contain the answer to the question, answer
with the following format

### Response

I do not know.

### Input

question_type:  factor

question:  What can amd convert to?

context:  Explained to patient that he does have AMD but it is mild
and
```

### E.2.2 EXAMPLES OF WRONG PREDICTIONS

Listing 10: OPT-IML-MAX-30B with guided instruction-based prompting: wrong prediction

```
You are a helpful medical knowledge extractor assistant. Your task is to
    answer to the question, using only and only the provided context. Do
    not use any other sources. Do not use your knowledge. Answer to the
    question using the context. Do not write explanation. Do not write
    long sentences. Answer with extracted entities, separated with commas.
     I will provide the inputs in the following format

### Input
question_type: treat/coexists_with/factor
question: [QUESTION]
context: [CONTEXT]

I want you to answer to the question in the following format, depending
    on the [ENTITY_TYPE] which can be either [TREAT], [COEXISTS_WITH], [
    FACTOR]. Do not include other entity types. Do not include treatment
    if not asked. Only extract entities for that type.

### Response

[ENTITY_TYPE]: [ENTITY_1], [ENTITY_2], [ENTITY_3]...

If the question is not related to the context, or if the context contains
     a missing information to answer to the question, answer "I do not
    know." and nothing more. No explanations. If the context does not
    contain the answer to the question, answer with the following format

### Response
I do not know.

### Input

question_type: coexists_with
question: What can armd convert to?
context: Dry ARMD - Explained that there is no specific treatment at this
     time. Can convert to wet ARMD, which is treatable. Eating a healthy
    diet that includes green leafy vegetables, fish, taking AREDS + WACS
    eye vitamins and Omega-3 fatty acids has been shown to significantly
    decrease the progression of the disease. Monitor closely, daily use
    of amsler grid. All questions were answered to the patient's
    satisfaction

### Response
I do not know
```

Listing 11: OPT-IML-MAX-30B with guided instruction-based prompting: wrong prediction

```
You are a helpful medical knowledge extractor assistant. Your task is to
    answer to the question, using only and only the provided context. Do
    not use any other sources. Do not use your knowledge. Answer to the
    question using the context. Do not write explanation. Do not write
    long sentences. Answer with extracted entities, separated with commas.
     I will provide the inputs in the following format

### Input
question_type: treat/coexists_with/factor
question: [QUESTION]
context: [CONTEXT]

I want you to answer to the question in the following format, depending
    on the [ENTITY_TYPE] which can be either [TREAT], [COEXISTS_WITH], [
    FACTOR]. Do not include other entity types. Do not include treatment
    if not asked. Only extract entities for that type.

### Response

[ENTITY_TYPE]: [ENTITY_1], [ENTITY_2], [ENTITY_3]...

If the question is not related to the context, or if the context contains
     a missing information to answer to the question, answer "I do not
    know." and nothing more. No explanations. If the context does not
    contain the answer to the question, answer with the following format

### Response
I do not know.

### Input

question_type: coexists_with
question: What does armd lead to?
context: Acute Exudative ARMD OU - chronic leakage OU. No significant
    changes OU. Recommend treating with Anti-VEGF injections. Discussed
    with patient the gravity of this disease, including the potential for
     vision loss and guarded recovery. Reviewed treatment options -
    Avastin, Lucentis, Eylea, PDT.

### Response
I do not know
```

Listing 12: Vicuna-33B with instruction-based prompting: wrong prediction

```
Instruction: I want you to act as a medical question answering machine. I
     will provide you with
questions and a context, and you will reply with the answers. If the
    answer is not in context, answer "I do not know."
question: What can decrease the chance of armd?
context: Acute Exudative ARMD OS - Subertinal blood/exudates OU-Discussed
     with patient the gravity of this disease, including the potential
    for vision loss and guarded recovery. Reviewed treatment options -
    Avastin, Lucentis, Eylea, PDT.

Answer:
In the context provided, the question is about decreasing the chance of
    ARMD (Age-Related Macular Degeneration).  While I am not a doctor, I
    can provide some general information on this topic.There are certain
    risk factors for ARMD that you may not be able to change, such as age,
    family history, and genetics.  However, there are some lifestyle
    changes that may help decrease the risk of ARMD or slow its
    progression:Maintain a healthy diet rich in fruits, vegetables, and
    whole grains.  Foods high in antioxidants, such as vitamins C and E,
```

```
      zinc, and lutein, may be particularly beneficial.Exercise regularly,
      as studies have shown that physical activity may reduce the risk of
      ARMD.Maintain a healthy weight, as obesity has been linked to a higher
      risk of ARMD.Quit smoking, as smoking is a risk factor for ARMD and
      can exacerbate the condition.Regularly check your vision and have
      annual eye exams, especially if you have a family history of ARMD or
      other risk factors.  Early detection and treatment can help manage
      the condition more effectively.It is important to discuss these
      options with a healthcare professional to determine the best course
      of action for your specific situation.
```

Listing 13: LLama-2-70B-chat with instruction-based prompting: wrong prediction

```
Instruction: I want you to act as a medical question answering machine. I
     will provide you with
questions and a context, and you will reply with the answers. If the
    answer is not in context, answer "I do not know."
question: What can heal amd? # TODO change heal to our question
context: Wet ARMD (Inactive) OD - Improved, no active leakage was seen
    during dilated eye exam and diagnostic testing. Patient is advised to
     continue AREDS + WACS vitamins. Eat green leafy vegetables like
    Spinach 5 times a week and fish at least 2 times a week.Recommend
    close monitoring and treatment as indicated. Patient will return to
    office in 1 month for IVFA to monitor AMD.
Answer:
I cannot provide medical advice or recommend treatments for specific
    individuals.  However, I can provide some general information about
    age-related macular degeneration (AMD) and its treatment options.AMD
    is a chronic eye condition that affects the macula, the part of the
    retina responsible for central vision.  There are two main types of
    AMD: dry and wet.  Wet AMD is characterized by the growth of new
    blood vessels under the retina, which can leak fluid and cause vision
    loss.While there is no cure for AMD, there are several treatment
    options available to slow down its progression and manage its
    symptoms.  For wet AMD, anti-vascular endothelial growth factor (VEGF)
    injections are commonly used to prevent further vision loss.  These
    medications work by blocking the growth of new blood vessels and
    reducing the amount of fluid in the retina.In addition to medication,
    a healthy lifestyle can also help manage AMD. A diet rich in fruits,
    vegetables, and omega-3 fatty acids, along with regular exercise and
    not smoking, can reduce the risk of AMD progression.It's important to
    note that the best course of treatment for AMD will depend on the
    individual patient and the severity of their condition.  Regular eye
    exams and consultations with an eye care professional are necessary
    to determine the most appropriate treatment plan.In the context of
    the patient you described, the recommended treatment plan includes
    continuing AREDS + WACS vitamins, eating a healthy diet, and close
    monitoring with regular eye exams.  IVFA (intravitreal fusion) may
    also be recommended to monitor the progression of AMD and determine
    if additional treatment is needed.  However, it's important to
    consult with an eye care professional for personalized medical
    advice.
```

Listing 14: LLama-2-70B-chat with instruction-based prompting: wrong prediction

```
Instruction: I want you to act as a medical question answering machine. I
     will provide you with
questions and a context, and you will reply with the answers. If the
    answer is not in context, answer "I do not know."
question: What can reduce the risk of armd?
context: Acute Exudative ARMD OS - ICG/IVFA shows active bleeding.
    Discussed with patient the gravity of this disease, including the
    potential for vision loss and guarded recovery. Reviewed treatment
    options - Avastin, Lucentis, Eylea, PDT.
Answer:
I do not know.The question asked is not in context with the given
    situation.  The patient has acute exudative age-related macular
    degeneration (ARMD) with active bleeding, and the doctor has already
    discussed the severity of the disease and the potential treatment
    options with the patient.  The question about what can reduce the
    risk of ARMD is not relevant to the current situation.
```

### E.2.3 EXAMPLES OF RIGHT PREDICTIONS

Listing 15: WizardLM-70B with guided instruction-based prompting: right prediction

```
You are a helpful medical knowledge extractor assistant. Your task is to
    answer to the question, using only and only the provided context. Do
    not use any other sources. Do not use your knowledge. Answer to the
    question using the context. Do not write explanation. Do not write
    long sentences. Answer with extracted entities, separated with commas.
     I will provide the inputs in the following format

### Input
question_type: treat/coexists_with/factor
question: [QUESTION]
context: [CONTEXT]

I want you to answer to the question in the following format, depending
    on the [ENTITY_TYPE] which can be either [TREAT], [COEXISTS_WITH], [
    FACTOR]. Do not include other entity types. Do not include treatment
    if not asked. Only extract entities for that type.

### Response

[ENTITY_TYPE]: [ENTITY_1], [ENTITY_2], [ENTITY_3]...

If the question is not related to the context, or if the context contains
     a missing information to answer to the question, answer "I do not
    know." and nothing more. No explanations. If the context does not
    contain the answer to the question, answer with the following format

### Response
I do not know.

### Input

question_type: factor
question: What can increase the risk of armd?
context: 2 small Druse OD- clinically does not look like ARMD. Patient
    has a family history of ARMD, recommend starting on AREDS + WACS
    vitamins. Eat green leafy vegetables like Spinach 5 times a week and
    fish at least 2 times a week.

### Response
factor:  factor:  family history, diet, lifestyle
```

Listing 16: Vicuna-33B with guided instruction-based prompting: right prediction

```
You are a helpful medical knowledge extractor assistant. Your task is to
    answer to the question, using only and only the provided context. Do
    not use any other sources. Do not use your knowledge. Answer to the
    question using the context. Do not write explanation. Do not write
    long sentences. Answer with extracted entities, separated with commas.
     I will provide the inputs in the following format

### Input
question_type: treat/coexists_with/factor
question: [QUESTION]
context: [CONTEXT]

I want you to answer to the question in the following format, depending
    on the [ENTITY_TYPE] which can be either [TREAT], [COEXISTS_WITH], [
    FACTOR]. Do not include other entity types. Do not include treatment
    if not asked. Only extract entities for that type.
```

```
### Response

[ENTITY_TYPE]: [ENTITY_1], [ENTITY_2], [ENTITY_3]...

If the question is not related to the context, or if the context contains
    a missing information to answer to the question, answer "I do not
    know." and nothing more. No explanations. If the context does not
    contain the answer to the question, answer with the following format

### Response
I do not know.

### Input

question_type: treat
question: What can decrease the chance of armd?
context: Acute Exudative ARMD OS - Subertinal blood/exudates OU-Discussed
    with patient the gravity of this disease, including the potential
    for vision loss and guarded recovery. Reviewed treatment options -
    Avastin, Lucentis, Eylea, PDT.

### Response
treat:  Avastin, Lucentis, Eylea, PDT
```

Listing 17: LLama-2-70B-chat with guided instruction-based prompting: right prediction

```
You are a helpful medical knowledge extractor assistant. Your task is to
    answer to the question, using only and only the provided context. Do
    not use any other sources. Do not use your knowledge. Answer to the
    question using the context. Do not write explanation. Do not write
    long sentences. Answer with extracted entities, separated with commas.
     I will provide the inputs in the following format

### Input
question_type: treat/coexists_with/factor
question: [QUESTION]
context: [CONTEXT]

I want you to answer to the question in the following format, depending
    on the [ENTITY_TYPE] which can be either [TREAT], [COEXISTS_WITH], [
    FACTOR]. Do not include other entity types. Do not include treatment
    if not asked. Only extract entities for that type.

### Response

[ENTITY_TYPE]: [ENTITY_1], [ENTITY_2], [ENTITY_3]...

If the question is not related to the context, or if the context contains
     a missing information to answer to the question, answer "I do not
    know." and nothing more. No explanations. If the context does not
    contain the answer to the question, answer with the following format

### Response
I do not know.

### Input

question_type: treat
question: What can slow the progression of armd?
context: Dry ARMD OU- Explained that there is no specific treatment at
    this time. Can convert to wet ARMD, which is treatable. Eating a
    healthy diet that includes green leafy vegetables, fish, taking AREDS
     + WACS eye vitamins and Omega-3 fatty acids has been shown to
```

```
    significantly decrease the progression of the disease. Monitor
    closely. All questions were answered to the patient's satisfaction.

### Response
treat:  AREDS + WACS eye vitamins, Omega-3 fatty acids, healthy diet
    including green leafy vegetables, fish
```

## E.3 ENCODER-DECODER MODELS

### E.3.1 EXAMPLES OF WRONG PREDICTIONS

Listing 18: FLAN-UL2 with instruction-based few-shot prompting: wrong prediction

```
Instruction: I want you to act as a question answering machine. I will
    provide you with a question and a context, and you will reply with
    the answers.
Question: What can slow the progression of AMD?
Context: Macular Dystrophy vs. Early Dry AMD OU - Explained that there is
     no specific treatment at this time. Patient educated on condition.
    Eating a healthy diet that includes green leafy vegetables, fish,
    taking AREDS + WACS eye vitamins and Omega-3 fatty acids has been
    shown to significantly decrease the progression of the disease.
Answer: Eating a healthy diet that includes green leafy vegetables.

Question: What can slow the progression of myopic macular degeneration?
Context: Myopic Macular Degeneration - Explained that there is no
    specific treatment at this time. AREDS and WACS vitamins MAY help
    slow down the progression of the degeneration. Monitor closely. All
    questions were answered to the patients satisfaction.
Answer: AREDS and WACS vitamins

Question: What can myopic macular degeneration convert to?
Context: Myopic Macular Degeneration - Explained that there is no
    specific treatment at this time. AREDS and WACS vitamins MAY help
    slow down the progression of the degeneration. Monitor closely. All
    questions were answered to the patients satisfaction.
Answer: AREDS + WACS eye vitamins
```

### E.3.2 EXAMPLES OF RIGHT PREDICTIONS

Listing 19: FLAN-T5-XXL with instruction-based prompting: right prediction

```
I want you to act as a question answering machine. I will provide you
    with a questions and a context and you will reply with the answers.
Question: What can slow the progression of armd?
Instruction: If the answer is not in context answer "i do not know".
Context: Wet ARMD- Patient educated on condition. Discussed diet and use
    of Macular Degeneration vitamins. All questions were answered to the
    patient's satisfaction.
Answer:
vitamins
```

Listing 20: FLAN-T5-XXL with few-shot prompting: right prediction

```
question: What can slow the progression of macular disease?
context: very strong association between smoking and macular disease.
    Smokers are up to four times more likely than non-smokers to develop
    age-related macular degeneration (AMD) and smokers with a genetic
    predisposition to AMD are eight times more likely to get the
    condition. Some research suggests people with certain genes can be 20
     times more likely to get macular degeneration if they smoke.
target: the answer to the question given the context is smoking.
```

```
question: What can slow the progression of amd?
context: Macular Dystrophy vs. Early Dry AMD OU - Explained that there is
    no specific treatment at this time. Patient educated on condition.
    Eating a healthy diet that includes green leafy vegetables, fish,
    taking AREDS + WACS eye vitamins and Omega-3 fatty acids has been
    shown to significantly decrease the progression of the disease.
    Stressed the need for follow up exams. All questions were answered to
    the patient's satisfaction.
target: the answer to the question given the context is Eating a healthy
    diet that includes green leafy vegetables.

question: What can slow the progression of myopic macular degeneration?
context: D/w pt: Myopic macular degeneration. Diagnosis discussed with
    patient. Possible treatments explained including glasses, refractive
    surgery, contact lenses or doing nothing. All questions were answered
    to patients satisfaction.
target: the answer to the question given the context is glasses
```

Listing 21: FLAN-UL2 with few-shot prompting: right prediction

```
very strong association between smoking and macular disease. Smokers are
    up to four times more likely than non-smokers to develop age-related
    macular degeneration (AMD) and smokers with a genetic predisposition
    to AMD are eight times more likely to get the condition. Some
    research suggests people with certain genes can be 20 times more
    likely to get macular degeneration if they smoke.
Create a bulleted list of what can slow the progression of macular
    disease?
- not smoking

Macular Dystrophy vs. Early Dry AMD OU - Explained that there is no
    specific treatment at this time. Patient educated on condition.
    Eating a healthy diet that includes green leafy vegetables, fish,
    taking AREDS + WACS eye vitamins and Omega-3 fatty acids has been
    shown to significantly decrease the progression of the disease.
    Stressed the need for follow up exams. All questions were answered to
    the patient's satisfaction. target: the answer to the question given
    the context is Eating a healthy diet that includes green leafy
    vegetables.
Create a bulleted list of what can slow the progression of amd?
- Eating a healthy diet
- Green leafy vegetables

Myopic Macular Degeneration - Explained that there is no specific
    treatment at this time. AREDS and WACS vitamins MAY help slow down
    the progression of the degeneration. Monitor closely. All questions
    were answered to the patient's satisfacti
Create a bulleted list of What can slow the progression of myopic macular
    degeneration?.
- AREDS
- WACS vitamins
```

Listing 22: FLAN-UL2 with instruction prompting: right prediction

```
Instruction: I want you to act as a medical question answering machine. I
    will provide you with
questions and a context, and you will reply with the answers.
Question: What does armd affect?
Instruction: If the answer is not in context, answer "I do not know."
Context: Acute Exudative ARMD/ CSCR OD - appears slightly worse on OCT
    and exam. Reviewed treatment options - Avastin, Lucentis, Eylea, PDT.
Answer: I do not know
```

# F    TECHNICAL DETAILS

| Hyperparameter | Value |
|---|---|
| threshold_preprocessing | 0.8 |
| threshold_notes_identification | 0.8 |
| relation_occurrence_number | 10 |
| relation_probability | 0.1 |

Table 6: Hyperparameters of the system. They may be tuned for a dataset.

