# OpenReview forum: "LARGE LANGUAGE MODELS FOR BIOMEDICAL KNOWLEDGE GRAPH CONSTRUCTION"
_ICLR.cc/2024/Conference — ICLR 2024 Conference Withdrawn Submission_

### Official Review · Reviewer_ycCD · 2023-10-20

**Soundness:** 2 fair
**Presentation:** 2 fair
**Contribution:** 1 poor
**Rating:** 1
**Confidence:** 5

**Summary:**

This study explores using large language models (LLMs) to automatically construct knowledge graphs (KGs) in medicine for applications such as drug discovery and clinical trial design. The research evaluates 12 different LLMs, considering their performance and safety attributes, using electronic medical record notes to identify medical entities like diseases, factors, treatments, and manifestations. Results indicate that decoder-only LLMs require further investigation, and the methodology is applied to age-related macular degeneration.

**Strengths:**

One notable advantage of this study is its thorough investigation into the utility of large language models (LLMs) for the automated creation of knowledge graphs within the medical domain. The research offers a robust assessment of 12 diverse LLMs, shedding light on their performance and safety characteristics. Furthermore, the study's emphasis on real-world medical applications, including drug discovery and clinical trial design, underscores its practical significance in advancing healthcare-related research and technological advancements.

**Weaknesses:**

The contribution of this study is confusing. The authors claims that they evaluate the performance of various LLM for better KG construction. However, the metrics is used to evaluate QA performance.

The relation between the prompts and KG construction is demonstrated to provide convincing evidence to support the contribution argument.

There is neither science illustration about how to design the prompt nor reference from previous study to show the prompt design process.

The study is conducted with little science rigor.

The experimental design is not completely illustrated.

KG construction process is not clearly articulate.

**Questions:**

Could you share more information about LLM-based KG given the public dataset?

How to evaluate the quality of KG construction?

Why use QA performance to demonstrate KG construction performance?

How did you construct your KG? Just by using the prompts?

Does prompts impact the model performance?

---

> ### Author Response · Authors · 2023-11-16
> **Response**
>
> Thank you for your appreciation of the comprehensive analysis of the performance and safety attributes of LLMs with varying architectures.
>
> We are providing an end-to-end ML framework to construct KGs from the EMR notes. It includes several steps as presented in the 4th section:
> 1. Preprocessing
> 2. Disease-specific notes identification
> 3. Prompt design, which is provided under section 4.4, where we show that we use a systematic and task-agnostic process to construct prompts as outlined in Jimenez Gutierrez et al. (2022).
> 4. Question design, the questions are categorized into three types: treatment-related,
> factor-related, and coexists with related questions.
> 5. Relation extraction
> 6. Postprocessing
>
> To evaluate the LLMs, we needed to review all EMR notes and extract all factors, treatments, and coexisting terms. The evaluation is based on precision and recall, with the ground truth for comparison being the entity values available in the notes. The precision and recall we calculate provide an evaluation of the whole framework, thus the KG, not only the QA system. With these metrics, we can compare the relations our framework gives as an output to the ones extracted by medical practitioners.  We had a total of 360,000 EMR notes. To avoid excessive time consumption, we chose to conduct the final analysis only on notes related to one disease (age-related macula degeneration), thereby annotating only age-related macula degeneration notes. In the results section, we provide precision and recall results for all LLMs, besides some of the decoder-only models, with which it was not possible to get structured output essential for the KG. Thus we give separate analyses for those decoder-only models. Afterward, we concentrated on qualitative analysis in the last part of the results section comparing the results with the relations extracted by the SemMedDB, which is a repository of semantic predictions extracted from the titles and abstracts of all PubMed citations.
>
> Answers to the questions
>
> 1) The dataset we used is a real-world EMR notes provided by the Macula & Retina Institute. You can find some examples in the Appendix section. The decision to use this dataset was to ensure that none of the LLMs have been trained or fine-tuned on it, ensuring a fair comparison. At the end of the "Related Work" section, we describe why this dataset poses greater challenges compared to those in other papers. Thank you for the suggestion, we will consider using a public dataset for the evaluation of our method in future versions of our papers, but also before that we should ensure that none of the models have been trained/or fine-tuned on that.
>
> 2) You can find about the evaluation of the KG in the aforementioned text.
>
> 3) We are providing an evaluation of the KG construction process both quantitatively and qualitatively, but also give some examples of the prompt results in the Appendix to show how the prompt design works qualitatively.
>
> 4) You can find the KG construction process under the 4th section.
>
> 5) Yes, prompts impact the KG construction process as can be visible in Table 2. The results are different by using 0-shot, few-shot, instruction-based, and our demonstrated "guided" prompt.

---

### Official Review · Reviewer_wiD2 · 2023-10-29

**Soundness:** 3 good
**Presentation:** 2 fair
**Contribution:** 2 fair
**Rating:** 5
**Confidence:** 3

**Summary:**

Authors focus on the research on biomedical knowledge graph (KG) automatic construction in their study, where they propose an end-to-end machine learning solution based on large language models (LLMs) that utilize electronic medical record notes to construct KGs. In their study, the entities are selected to diseases, factors, treatments, as well as manifestations. In terms of the LLMs, they embark on an assessment of 12 LLMs of various architectures. Based on their observations on these 12 LLMs, they conclude that in contrast to encoder-only and encoder-decoder, decoder-only LLMs require further investigation.

**Strengths:**

Authors carry out a good work on the automatic construction of biomedical knowledge graph by using Large Language Model. This research provides a new technique framework to researchers in knowledge graph.  Also, authors clearly demonstrate their instruction-based prompts that make most LLMs work.

**Weaknesses:**

Nevertheless, I have some concerns to this study.
1. Authors take a preprocessing operation for clinic notes. So, how to ensure its reasonableness. It would be better to demonstrate the different performance of LLMs on processed data and raw data, respectively.
2. This study is mainly related to an assessment work for different LLMs; however, authors only evaluate the precision and recall on three medical entities in Tab. 5 (I think it should be Tab.2) besides demonstrating some cases to prove their observations. It would be better to discuss different evaluations for different LLMs, such as reasons of hallucination, accuracy and evidence to extracted triplets, and so on.
3. In Tab.2, there have no explicit results to show the effectiveness of guide instruction-based prompting. It would be better to show the performance of “Vicuna-33B w/o guide, Llama-2-70B w/o guide and WizardLM-70B w/o guide”.
4. Why do not evaluate GPT-4? Could authors carry out an experiment based on GPT-4?
5. The presentations should be improved.
For example:
(1) “Tab.5” in “Precision and Recall Results” should be “Tab.2”;
(2) What does “Section ? 0.8” in “DATASET PREPROCESSING” refer to?
(3) Why put the “hallucination” in Appendix?
(4) What does the concepts of “Coexists with, Factor, Treatment” represent? “question types”, “medical entities”, or “relations”.
……

**Questions:**

See weaknesses.

---

> ### Author Response · Authors · 2023-11-16
> **Response**
>
> Thanks a lot for the comments.
>
> 1) We will add that comparison in the next version of our paper.
>
> 2) accuracy and evidence to extracted triplets: such analysis is provided in the "Qualitative Example: AMD" part of the Results section. We analyze each relation by bringing forward scientific literature proving their truthfulness.
> hallucination: usually it is hard to analyze the reason for the hallucinations of LLMs, it may be related to the domain shift between the training and testing data. We will consider finetuning models in such data and experimenting with them in future versions of our paper.
>
> 3) Quantitative results for decoder-only models (including Vicuna-33B, Llama-2-70B, and WizardLM-70B ) using 0-shot, few-shot, and instruction-based prompting techniques are not available. These models (with any of these three prompting techniques) did not produce structured outputs, rendering them unsuitable for our task. Guided instruction-based prompting has shown significant improvements for the 3 models provided in Table 2. So, there are no quantitative results without guided prompt design for those three models.
>
> 4) We are using a private dataset. We have not experimented with any of the GPTs for data privacy reasons.
>
> 5.1) Changed
> 5.2) You can find clarification in the new version
> 5.3) All of the examples (including hallucinations) are quite long, so we have decided to put them in the Appendix
> 5.4) The explanations are presented in the 4.5 section. Here they are:
>
> a) The treatment-related questions inquire about methods to slow down the progression, decrease the chance, or reduce the risk of a specific condition.
> b) The factor-related questions aim to identify the causes, factors, or risks associated with a condition.
> c) The coexists with-related questions explore any symptoms, effects, diseases, clinical tests, or behaviors that may manifest in the patient while experiencing the disease

---

> ### Comment · Reviewer_wiD2 · 2023-11-18
>
> Thanks so much for your consideration of my suggestions and giving me responses about my questions. Nevertheless, I still would like to discuss some questions with the author.
> 1. I think the “Qualitative Example: AMD” you mentioned here is an application discussion rather than an accuracy analysis. I think the accuracy discussion should face to the triplet extraction. For example, show the precent of LLMs in extracting a totally correct triplet. From Tab.2 in your manuscript, I cannot gain the performance of LLMs in extracting a KG triplet, i.e., (head entity, relation, tail entity) or (head entity, tail entity, relation), although you may have done this work.
> 2. I agree with your opinion in LLM’s hallucination. But I also think we can give readers a possible analysis in a specific question. For example, maybe the inputs of LLM are confusion when we don’t use the instruction-prompt, etc.
> 3. Your response about my question in performance of guide instruction-based prompting is clear. However, I still believe it would be better to add a suitable ablation experiment on guide instruction-based prompting, because the results of Coexists with in Tab.2 are not better than encoder-decoder LLMs.
> 4. As I know, we can use Azure OpenAI now, so that we can test our data based on the ChatGPT’s API.
> I am very looking forward to your study in automatic knowledge graph construction.

---

> ### Author Response · Authors · 2023-11-21
> **Response.**
>
> Thanks a lot for the comprehensive analysis of our paper.  We appreciate your suggestions. Here are the answers to the questions:
>
> 1. The correlation between the performance assessment outlined in Table 2 and triple extraction results are following:
>
> Each column in the table corresponds to a specific relation type, such as 'treatment.' Precision represented as 95%, and recall, represented as 90%, indicate that, within the specified method, the model successfully extracts 90% of triplets (head entity, relation = treatment, tail entity) from the available data, and 95% of these extracted triplets are accurate.  The ground truth for comparison is the triplets available in the notes. All the available triplets have been extracted by medical practitioners manually.
> Additionally, reconstructing the Knowledge Graph (KG) is facilitated by connecting head entities to tail entities using the corresponding relation edges.
>
> This structured methodology allows for logical inferences on the KG. For example, if entity A treats entity B, and B is a factor for C, one can infer that A treats C as well. The "Qualitative Example: AMD" section presents all extracted triplets where the tail entity is AMD. Entities highlighted in red within the table signify inaccuracies in the triplets (head entity = red-colored entity, relation, tail entity = AMD), while an orange-colored triplet indicates the model's failure to extract a triplet present in the data (clinical notes).
>
> 2. We have faced such things during the design of our proposed "guided" instruction-based prompting.  I agree with your opinion, we will add the evolution of the prompt from standard to our design and give an analysis of what may cause confusion of the LLM to hallucinate.
>
> 3. The success metric for us when evaluating "guided" prompt design was not to achieve better results than encoder-decoder results, but to make decoder-only models to give enough structured outputs which is essential for relation extraction. Our hypothesis is that with finetuning on health-related data and with our prompt design they may perform much better. But for now, they have at least started to give responses at some structured level, thus pushing the performance of some of the decoder-only models from 0's to the numbers available in Table 2. Your suggestion on ablation studies is definitely valuable and we will consider it adding to our next version of the paper.
>
> 4. We have evaluated ChatGPT for the KG construction for AMD, it gave a very small number of relations compared to our results by using the clinical notes. We will add a comparison with ChatGPT to our paper.
>
> I hope my answers are helpful to better understand our approach. Please let us know if you have additional comments or suggestions.

---

### Official Review · Reviewer_a8TG · 2023-10-30

**Soundness:** 2 fair
**Presentation:** 1 poor
**Contribution:** 2 fair
**Rating:** 3
**Confidence:** 5

**Summary:**

In this paper, the authors propose an end-to-end machine learning solution based on large language models (LLMs) that leverages electronic medical record notes to build KGs. To assess the performance and safety attributes of this approach, the study evaluates 12 LLMs with different architectures. The paper also introduces guided prompt design for such LLMs and demonstrates the methodology's application in age-related macular degeneration research.

**Strengths:**

1. The motivation is clear that the automatic construction of knowledge graphs (KGs) is an important research area in medicine, and LLMs could be helpful in this problem.

2. The experiments are conducted on 12 LLMs, which cover most of the available LLMs.

**Weaknesses:**

1. The paper's writing requires substantial improvement, as it contains numerous typos and format errors. Examples include:
- The citations used in the paper are in the wrong format.
- In sections 3.1 and 4.1, there is a non-existent reference to `"Section ?0.8".
- The section cross-references are not in a unified format. Some examples include "Section A", "Appendix Appendix D", "4.1", "Appendix subsubsection E.2.1".
- There is a "Table 4.2" in section 4.2. And "Table 5" in section 5 should actually refer to Table 2.
- The authors use past tense and present tense alternatively throughout the paper.
- There is a typo "Knowdledge" in the first paragraph of the introduction.
- In Algorithm 2, the symbol between "similarity_score" and "threshold" is not correctly shown.
- In section 4.6, the authors represent disease and the set of diseases as  $d$ and $D$. However, in section 4.1, they are referred as $c_{input}$ and $C$, respectively.
- In section 4.6, the authors "average over multiple occurrences of the relation of the type $t$ between $e$ and $d$." However, I believe the variable $e$ is never introduced in the paper.

2. It is unclear how the constructed knowledge graph is evaluated. The authors only mention that "the evaluation is done based on precision and recall", but the process of the evaluation is not addressed. What are the ground truth labels here?

3. In section 4.1, the authors mention using the BioBERT NER model to extract a list of diseases. However, an evaluation of its accuracy is missing. It's possible that the NER model could introduce some erroneous information or ignore any essential diseases.

4. In section 4.1, it is said that "If the similarity is above some threshold, we add the result to the result list." Here, the authors should explicitly mention the exact value of the threshold and discuss how it is determined.

5. In the right part of Figure 1, it is unclear which sentences are inputs and which sentences are LLM responses. For example, the paragraph "If the question is not related to the context, ..." is within the "###Response" block, but it seems to be an input instead.

6. For the prompt designs, it would be better for the authors to discuss the reasons why they have such designs and why these designs could help.

7. In section 4.6, it is inappropriate to say that "The numbers are chosen arbitrarily and they may be tuned for a dataset." The authors should study the effect of these hyperparameters and justify how they are chosen in the experiments to enhance reproducibility. Similarly, in section 4.7, the authors should also discuss why the threshold of the similarity score is set to 0.8.

**Questions:**

See the weaknesses above. One additional question:

1.  In section 4.3, the authors "explore two distinct approaches for aligning large language models to the task: open-book QA and in-context learning." According to the explanations in this section, "in the case of open-book QA, a query consists of a question and a context", however, it is also said that for in-context learning, "it extends the prompt to include both the question and the context." The two statements seem to be the same. Could the authors further explain the difference between the two approaches?

**Details Of Ethics Concerns:**

In the "Acknowledgements" section, the authors acknowledge the sources from which they acquired private datasets, and also mention the names of the people with whom they had discussions during the course of this project.

---

> ### Author Response · Authors · 2023-11-22
> **Response**
>
> Thanks a lot for the comments. Here are the answers:
>
> 1. Yes, we agree. We addressed all of these comments in the latest version.
> 2.  The ground truth for comparison is the triplets available in the notes. All the available triplets have been extracted by medical practitioners manually. More details are as follows:
>
> Each column in the table corresponds to a specific relation type, such as 'treatment.' Precision represented as 95%, and recall, represented as 90%, indicate that, within the specified method, the model successfully extracts 90% of triplets (head entity, relation = treatment, tail entity) from the available data, and 95% of these extracted triplets are accurate. Additionally, reconstructing the Knowledge Graph (KG) is facilitated by connecting head entities to tail entities using the corresponding relation edges.
>
> This structured methodology allows for logical inferences on the KG. For example, if entity A treats entity B, and B is a factor for C, one can infer that A treats C as well. The "Qualitative Example: AMD" section presents all extracted triplets where the tail entity is AMD. Entities highlighted in red within the table signify inaccuracies in the triplets (head entity = red-colored entity, relation, tail entity = AMD), while an orange-colored triplet indicates the model's failure to extract a triplet present in the data (clinical notes).
>
> 3. We will add the evaluation of BioBERT NER in the next version of our paper.
>
> 4. Addressed in the latest version.
>
> 5. Addressed in the latest version.
>
> 6. For instruction-based prompting, we follow a systematic and task-agnostic process to construct prompts as outlined in Jimenez Gutierrez et al. (2022). As depicted in the left examples in Figure 1, this method identifies three key components of a prompt: overall task instructions, a sentence introduction, and a retrieval message. Building upon this, we introduce guided instruction-based prompting, denoted as 'guided' in the forthcoming sections. This refined prompt structure incorporates three fundamental elements:
>
> a) Contextual Task Instruction: This aspect furnishes explicit and comprehensive guidance, emphasizing the model's role in extracting information solely from the provided context. It establishes a clear framework for understanding the task at hand.
>
> b) Input Format Guidance: To mitigate issues of relations extracted out of context, we introduced explicit instructions on how the model should interpret and process input. This includes specifying the acceptable types and formats for questions and contexts.
>
> c) Output Format Guidance: We refined the retrieval message to include the entity type and explicitly instruct the model on the desired format for its responses.
>
> 7. We agree. Addressed in the latest version.
>
> 8. Thank you. We have given a more detailed description in the paper. Here is the summary:
>
> a) Open-Book QA: In this approach, the context serves as a specific, external information source from which the model is expected to extract answers to the posed questions. The context is typically a passage, document, or a set of information separate from the model's pre-existing knowledge. The model's task is to directly reference and pull information from this provided context to respond to the query. Thus, the context in open-book QA acts as a discrete reference material that the model consults to find answers.
>
> b) In-Context Learning: While in-context learning also involves a question and a context, the usage and purpose of the context differ significantly. Here, the context is not an external source of information but an integral part of the model's prompt. It includes a series of input-output pairs that serve as examples to guide the model in understanding the nature of the task. These examples demonstrate how to process and respond to similar queries. Therefore, the context in in-context learning is instructional and part of the learning material embedded within the prompt, guiding the model's response generation process based on demonstrated patterns, rather than serving as a source from which to extract direct answers.
>
> In summary, while both approaches use a combination of questions and context, the role of the context in open-book QA is as an external reference source, whereas in in-context learning, it is an intrinsic part of the learning prompt, providing instructional examples rather than factual content for direct query response.
>
> 9) Acknowledgements are removed in the latest version.
>
> We appreciate for such a detailed analysis of our paper. Please let us know if you have any additional comments.

---

### Official Review · Reviewer_7yEf · 2023-11-01

**Soundness:** 2 fair
**Presentation:** 3 good
**Contribution:** 1 poor
**Rating:** 3
**Confidence:** 3

**Summary:**

# Summary
LLMs for Biomedical KG Construction

## What is the problem?
Construction of biomedical knowledge graphs is challenging and involves a significant amount of expensive human expertise. This study examines whether or not they can be extracted automatically from clinical point-of-care notes.

## Why is it impactful?
Automatic construction of KGs would greatly accelerate the fields in biomedical science that rely on structured knowledge bases (of which there are many).

## Why is it technically challenging/interesting (e.g., why do naive approaches not work)?
Automatic KG construction relies on many techniques, including entity and relationship recognition, extraction, and normalization; parsing long documents in context; reasoning over clinical language; etc. All of these are rich areas of NLP that present longstanding research challenges.

## Why have existing approaches failed?
There are a host of existing approaches in the realm of automatic construction of KGs given LLMs. The authors reference almost none of these in their work, which reflects a major failing to contextualize their work in the context of related literature. These methods include the following:
  1. https://openreview.net/forum?id=ntIq8Wm79G-
  2. https://arxiv.org/pdf/2305.04676.pdf
  3. A number of papers from this list: https://github.com/zjukg/KG-LLM-Papers

## What is this paper's contribution?
  1. This paper releases a set of prompts that can help future researchers produce methods that automatically construct knowledge graphs given prompt-based LLM systems of various kinds.
  2. They also perform an evaluation of the efficacy of building KGs from text leveraging a variety of LLM architectures.

## How do these methods compare to prior works?
  1. There is insufficient comparison with prior works to say here. This is a major weakness of the work.

**Strengths:**

## Key Strengths (reasons I would advocate this paper be accepted)
  1. This is an important problem area.

## Minor Strengths (things I like, but wouldn't sway me on their own)
  2. Analyzing how differing pre-trained LLMs work in this setting is a useful analyses.
  3. The prompts released will likely be helpful.

**Weaknesses:**

## Key Weaknesses (reasons I would advocate this paper be rejected)
  1. You fail to sufficiently contextualize your advancement in the significant body of related literature in this space. There are a number of papers that complete or otherwise generate knowledge graphs via LLMs, and you don't reference or compare to these at all.
  2. You don't state that you plan to actually release the constructed knowledge graph. This seems a major limitation when such a graph would be a resource to the community, were it extracted in a meaningful way. While this may be because you are actually constructin per-patient KGs (and if so then this should be a minor weakness, not a key weakness) that is not sufficiently clearly stated in the text to alleviate this concern.
  3. If your KG is not patient specific, but is instead intended to capture general biomedical knowledge, it is not clear why using clinical point-of-care notes (as opposed to the clinical scientific literature) is the appropriate information source to use. Wouldn't scientific text be more accurate for capturing general information? If you are intending to capture per-patient information, why? For what will such information be used, and why should that be captured via unstructured clinical notes rather than structured information? These things need to be explained.
  4. Ultimately, the biggest weakness other than the lack of comparison to prior art is that this is likely just not well aligned with this venue. You do not have any significant methodological novelty here, and that is a key focus for ICLR. This is also not about representation learning, or really method development within ML at all, but is rather an (albeit important) application within NLP for health and biomedicine. Perhaps re-submitting to a venue more targeted to NLP contributions and applications (such as ACL, or possibly even better, EMNLP) would be more appropriate (though if the AC disagrees about the suitability I will of course retract this concern).

## Minor Weaknesses (things I dislike, but wouldn't sway me on their own)

**Questions:**

## What would make me raise my score? (Things that you can do that would, pending their results and the manner in which you accomplish them, make me raise my score)
I don't foresee a situation in which my score will raise unless the AC states that this is suitable for the venue. Even then, I would need to see major revisions to improve clarity and, critically, to contextualize this work amidst the related literature and quantify how this work makes novel methodological contributions for me to consider raising my score. Ideally such comparisons would include baselines from prior published methods to show that this method of KG construction is superior.

## Other Questions (things I'd like to know, but may or may not make me change my score)
None

---

> ### Author Response · Authors · 2023-11-16
> **Response**
>
> Thank you for the comments.
>
> 1) We will add them in the Related Work section.
>
> 2) The constructed KG is available under Table 3 in the "Qualitative Example: AMD" of the Results section.
>
> 3) In the "Qualitative Example: AMD" of the Results section we show that by using EMR unstructured notes we obtain information not available in the KG constructed by SemMedDB Kilicoglu et al. (2012). SemMedDB is a repository of semantic predictions extracted from the titles and abstracts of all PubMed citations. So, there is hidden valuable information in unstructured notes and extraction methods of this information are valuable for the scientific community and, more specifically can be used in the drug discovery and clinical trial design process.
>
> 4) The absence of absolute ground truth leaves us to compare the results with the knowledge of medical practitioners, which may be incomplete. Moreover, new knowledge is likely to be discovered when using large textual and EMR repositories Nordon et al. (2019). Thank you for your suggestions on future submissions. We will definitely consider them.

---

> > ### Comment · Reviewer_7yEf · 2023-11-22
> > **Thank you for your follow-up comments**
> >
> > Thank you for your response comments; especially in clarifying that Table 3 is your released KG, which I did not realize, as I expected your full KG to be much more expansive than that.
> >
> > Ultimately, I expect this will be an excellent submission to a more focused venue; good luck in all future submissions.

---

### Official Review · Reviewer_aBzH · 2023-11-03

**Soundness:** 1 poor
**Presentation:** 2 fair
**Contribution:** 2 fair
**Rating:** 3
**Confidence:** 4

**Summary:**

The paper provides an end-to-end approach  using LLMS for the construction of KGs from EMRs. it focuses on identifying notes in EMRs that are specific to certain diseases. To do this, they create template questions that look for specific types of information, such as treatments, contributing factors, and coexisting conditions. They experiment with different LLM structures - encoder-only, decoder-only, and encoder-decoder. They introduce guided instruction-based prompting to interact with LLMs.

**Strengths:**

Creating Knowledge Graphs automatically from Electronic Medical Record (EMR) notes is a significant challenge, and the approach introduced in the paper is original.

**Weaknesses:**

The paper's section on results looks at a very specific task  (age-related macula degeneration). Understanding how complex this dataset is compared to others would be really helpful. The paper also says that their method has precision 0.98 and recall of 1. But these scores are difficult to believe without comparing them to other standard results. The paper could be improved by giving more details about how this dataset stacks up against others, or by testing the method on different datasets that already exist. Right now, the paper isn't clear enough, and the reviewer has listed some questions that need clear answers to help understand the study better.

**Questions:**

(1) MIMIC-II is publicly available. But there are datasets like i2b2 which are carefully annotated by physicians and are subject to availability based on a license. Have you considered using this for evaluation?
(2) Can you explain why your method has recall 1 and precision 0.98 with FLAN-UL2?

---

> ### Author Response · Authors · 2023-11-16
> **Response**
>
> Thank you for your comments.
>
> 1) Thank you for the suggestion; we will consider using i2b2 for the evaluation of our method in future versions of our papers.
> 2) Our hypothesis is that prompt design and instructions specifically tailored for this use case aid LLMs in identifying all possible relations from the provided clinical notes.
> We provide guided prompt design for utilizing LLMs in this task.
>
> To evaluate the LLMs, we needed to review all EMR notes and extract all factors, treatments, and coexisting terms. The evaluation is based on precision and recall, with the ground truth for comparison being the entity values available in the notes. We had a total of 360,000 EMR notes. To avoid excessive time consumption, we chose to conduct the final analysis only on notes related to one disease (age-related macula degeneration), thereby annotating only age-related macula degeneration notes.
>
> One of the main contributions of our paper, besides presenting an end-to-end method leveraging LLMs for the automatic construction of KGs, is an extensive evaluation of the performance of 12 LLMs of various architectures specifically tailored for clinical relation extraction.
> The dataset we used is a real-world EMR notes provided by the Macula & Retina Institute. You can find some examples in the Appendix section. The decision to use this dataset was to ensure that none of the LLMs have been trained or fine-tuned on it, ensuring a fair comparison. At the end of the "Related Work" section, we describe why this dataset poses greater challenges compared to those in other papers.

---

### Meta-Review · Area_Chair_KSyV · 2023-12-05

**Metareview:**

This paper considers the problem of automatic Knowledge Graph (KG) construction in medicine using LLMs. Reviewers uniformly agreed that this is a potentially interesting and important direction.

However, the work fails to contextualize its contribution in the broader literature and existing methods; it is difficult as a result to appreciate the contribution on offer here. In addition, presentation issues further weaken the submission in its current form, and in particular hamper interpretation of the results reported.

**Justification For Why Not Higher Score:**

See comments above; primarily the issues here are (a) inadequate contextualization of the contribution w/r/t prior work, and, (b) poor presentation (as well as some issues with evaluations).

**Justification For Why Not Lower Score:**

N/A

---

### Decision · Program_Chairs · 2024-01-16

Reject